# Cytokine and phenotypic cell profiles in human cutaneous leishmaniasis caused by *Leishmania donovani*

**Hiruni Wijesooriya[◑], Nilakshi Samaranayake[◑]\*, Nadira D. Karunaweera[◑]**

Department of Parasitology, Faculty of Medicine, University of Colombo, Colombo, Sri Lanka

◑ These authors contributed equally to this work.
\* nilakshi@parasit.cmb.ac.lk

**Data Availability Statement:** All relevant data are within the paper and its Supporting information files or links to Figshare, which are referenced in the Results.

## Abstract

### Background

The innate immune mediators are likely to influence the clinical phenotype of leishmaniasis by primary responses which limit or facilitate the spread of the parasite, as well as by modulating adaptive immunity. This study investigated the response of key innate immune cells in a focus which regularly reports localised cutaneous leishmaniasis (LCL) caused by *Leishmania donovani*, a species which typically causes visceral disease.

### Methods

Peripheral blood mononuclear cell (PBMC) derived macrophages and dendritic cells from patients with LCL and healthy controls from endemic and non-endemic areas, were stimulated with soluble *Leishmania* antigen (SLA). Inflammatory mediators produced by macrophages (TNF-α/TGF-β/IL-10, ELISA; NO, Griess method) and dendritic cells (IL-12p70, IL-10, flowcytometry) and macrophage expression of surface markers of polarization, activation and maturation (flowcytometry) were determined at 24h, 48h and 72h and compared. Study was conducted prospectively from 2015–2019.

### Results

Patient derived macrophages and dendritic cells produced higher levels of both pro and anti-inflammatory mediators compared to controls ($p < 0.05$) with the best discrimination for active disease observed at 72h. Data demonstrated an early activation of macrophages and a subsequent pro-inflammatory bias, as indicated by temporal profiles of TNF-α/TGF-β and TNF-α/IL-10 ratios and higher proportions of classical (M1) macrophages. Higher TGF-β levels were observed in cells from patients with ulcerated or persistent lesions. Immune responses by cells derived from controls in endemic and non-endemic regions did not differ significantly from each other.

### Conclusions

The overall immunophenotypic profile suggests that LCL observed in the country is the result of a balancing immune response between pro-inflammatory and regulatory mediators.

**Funding:** This work was supported by the National Institute of Allergy and Infectious Diseases of the National Institutes of Health, USA, under award number U01AI136033 to NDK. The content is solely the responsibility of the authors and does not necessarily represent the official views of the National Institutes of Health. The funders had no role in study design, data collection and analysis, decision to publish, or preparation of the manuscript.

**Competing interests:** The authors have declared that no competing interests exist.

The mediators which showed distinct profiles in patients warrant further investigation as potential candidates for immunotherapeutic approaches. A comparison with visceral leishmaniasis caused by the same species, would provide further evidence on the differential role of these mediators in the resulting clinical phenotype.

## Introduction

Leishmaniasis is a vector-borne parasitic disease, caused by protozoan parasites of the genus *Leishmania*. The clinical picture of leishmaniasis is heterogeneous with a wide spectrum of human diseases, including cutaneous (CL), mucocutaneous and visceral (VL) forms. It is endemic in 98 countries with an estimated annual incidence of 30 000 cases of VL and more than 1 million cases of CL [1]. The cutaneous form of leishmaniasis itself has diverse presentations; the most common being localized cutaneous leishmaniasis (LCL). Further, in areas endemic for *Leishmania* transmission, approximately 10% of individuals may have evidence of exposure to parasite but lack disease pathology [2] and are classified as those with subclinical infections. Over twenty species of *Leishmania* species are recognized to infect humans. LCL is mainly caused by the species *L. tropica*, *L. aethiopica*, and *L. major* in the Old World. In the New World multiple species such as *L. amazonensis*, *L. infantum*, *L. mexicana*, *L. braziliensis*, *L. guyanensis*, *L. panamensis*, and *L. peruviana* cause LCL [3, 4].

The outcome of infection in leishmaniasis is complex, depending not only on the parasite species but also on the immune status of the host. The clear dichotomy of Th1/Th2 reactions seen in experimental models, with a Th1 type protective response being associated with localised cutaneous disease, is not always observed in leishmaniasis in human hosts. Evidence from more recent studies suggests that the innate immune response plays a pivotal role in the outcome of *Leishmania* infections. This response would act both in controlling parasite growth during the early stages of infection as well as in driving the cytokine microenvironment in which parasite-specific T cells are primed [5–8].

Macrophages and dendritic cells; two main cell types of the innate immune system, play a decisive role in the initial interactions between the parasites and the immune system of the host [9, 10]. Macrophages serve as host cells as well as the effector cells which finally eliminate the parasite, whereas both are accessory cells that present parasite antigens, deliver co-stimulatory signals and secrete cytokines modulating the subsequent adaptive immune response. Two subsets of macrophages are well recognized where classically activated (M1) macrophages are microbicidal while alternatively activated (M2) macrophages are anti-inflammatory. T helper cell differentiation is influenced by these antigen presenting cell subsets where M1 potentially induces Th1 and Th17 cells, while M2 induces Th2 and Treg cells [11].

It is noteworthy that macrophages generally lack the ability to induce the primary stimulation of specific T cells, which is carried out by dendritic cells, thus linking the innate and adaptive immune systems. The maturation status of dendritic cells and distinct dendritic cell subsets can induce different T helper cell responses and thereby modulate the subsequent immune responses to pathogens [12, 13]. Sri Lanka, an island nation in South Asia, mostly reports LCL with a few reports of mucosal [14] and visceral [15, 16] disease. The causative agent of all clinical disease types in Sri Lanka has been identified as *Leishmania donovani* MON 37 [17, 18].

*L. donovani* is usually associated with VL in the Old World and this species has been found to cause CL in only a few other foci; in Kenya [19], Yemen [20] and in the Himalayan region

of northern India [21]. The pathogenic mechanisms which limit this usually visceralizing species to localised cutaneous lesions remain largely unknown, with only limited reports from Sri Lanka on possible contributory host factors based on experimental models [22] and human studies [23–26].

The objective of this study was to characterize the innate cellular immune responses associated with locally acquired cutaneous leishmaniasis due to *L. donovani*. We hypothesized that distinct profiles of the early immune response determined the clinical phenotype of LCL observed in Sri Lanka. We aimed to investigate these immunological responses in LCL in relation to patients, non-endemic controls as well as endemic controls, with the latter likely to have been exposed but not developed symptomatic infections.

## Materials and methods

### Ethics approval and consent to participate

The study was conducted in compliance with the principles of Declaration of Helsinki [27]. All participants provided informed written consent and study protocols were approved by the Ethics Review Committee of Faculty of Medicine, University of Colombo, Colombo, Sri Lanka (EC-14-066).

### Study locations, participants and sample collection

This study was conducted prospectively during a period of 4-years (2015–2019). Altogether sixty patients with locally-acquired cutaneous leishmaniasis were recruited for the study from the leishmaniasis clinic held weekly at the Department of Parasitology, Faculty of Medicine, University of Colombo and dermatology clinics in selected Base Hospitals in endemic areas. The exclusion criteria consisted a history of inflammatory / immunosuppressive medical conditions. Details of the patients were collected using a pretested questionnaire by directly interviewing and examining the patients. Diagnosis was confirmed by direct microscopy and/ or culture of lesion material obtained by aspirates or slit skin smears. 5ml of peripheral venous blood was collected by venipuncture to EDTA tubes from each participant. All patients were routinely tested by rK39 immunochromatographic rapid diagnostic test to exclude serological evidence of visceral involvement.

Sixty endemic controls and thirty non-endemic controls, within the same age range and with the same gender distribution as the patients, with no history of inflammatory /immune suppressive medical conditions comprised the comparison group. Healthy non endemic controls were residents from areas which are not endemic for leishmaniasis in Sri Lanka. Healthy household contacts of diagnosed patients with no past history suggestive of leishmaniasis were recruited as endemic controls (Fig 1).

### Preparation of soluble *Leishmania* antigen (SLA)

*L. donovani* parasites were cultured in complete M199 (Gibco, USA) media (Hank's salts, L-Glutamine, 25mM HEPES, L-Amino Acids) with 5% heat inactivated fetal bovine serum (Sigma, USA) and 0.1% gentamycin. SLA was prepared from parasite isolates after 2 to 3 passages in liquid culture, as previously described [28]. Briefly, the promastigotes were harvested in late-log-phase and washed three times in 5 ml of cold sterile phosphate-buffered saline (PBS) followed by six cycles of freezing and thawing (frozen in liquid nitrogen and thawed at 37˚C). The resulting suspension was centrifuged at 2500 rpm at 4˚C for 5 min and the supernatant containing SLA was stored at -70˚C until use. The protein concentration in the supernatant was measured by Lowry's protein quantification method [29].

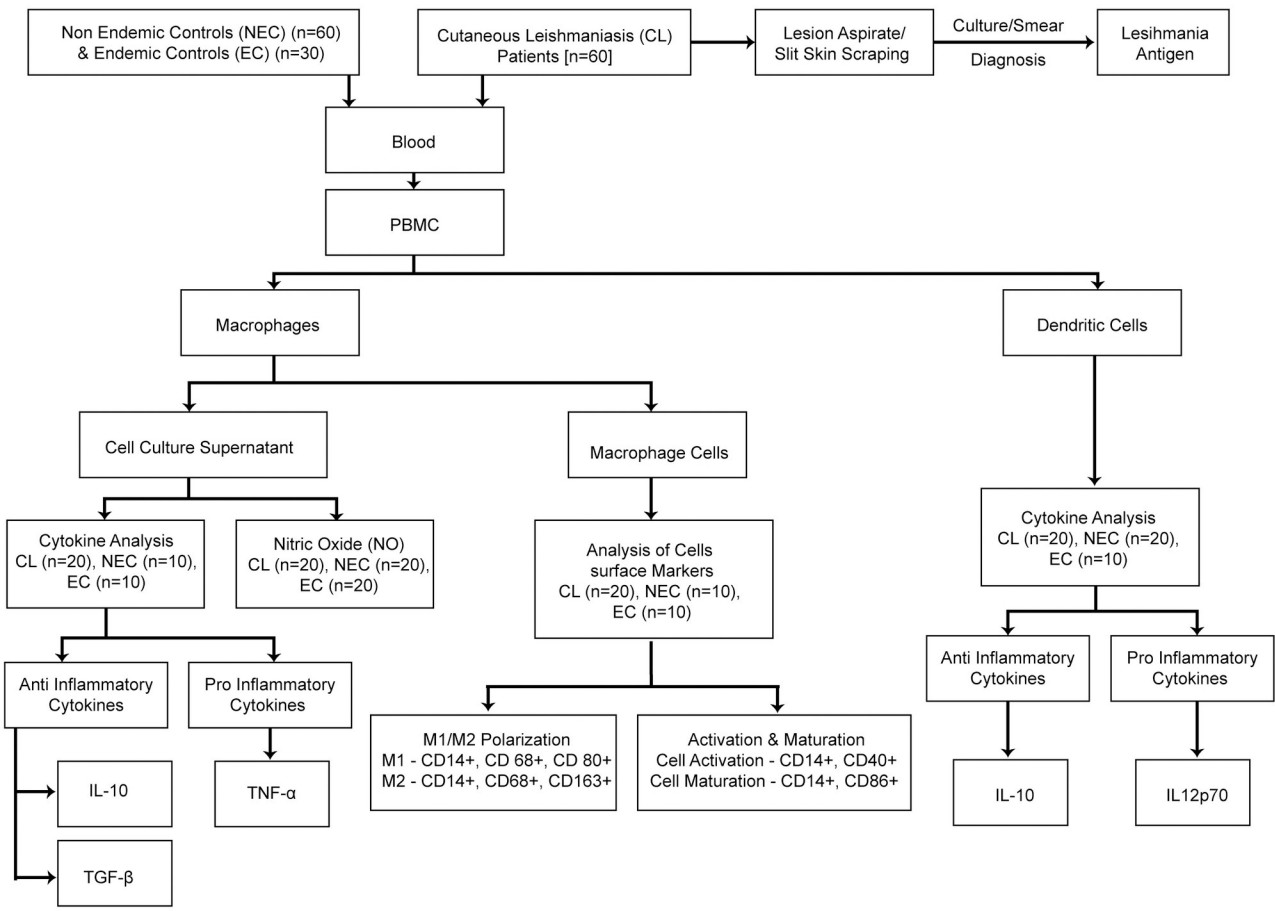

**Fig 1. Participant recruitment and sample analysis plan.**

## Generation of human monocyte-derived macrophages and dendritic cells *in-vitro*

PBMCwere isolated from EDTA blood by density gradient centrifugation using 30% of Ficoll-Paque (GE Life Sciences, country).

Cells were incubated at 37˚C in 5% $CO_2$ and monocytes were separated by adherence after 24 hours and cultured in complete RPMI1640 (Gibco, USA) supplemented with 5% human heat-inactivated serum (Sigma, USA), 10% of fetal calf serum (FCS) (Sigma, USA) and100 U/mL Penicillin-Streptomycin(Sigma, USA) in 24 well plates (1 X $10^6$cells/well) [30, 31]. The adherent cells displayed characteristics of monocyte derived macrophages after 5–6 days.

Dendritic cells were generated by culturing the monocytes separated by adherence at 24hrs in complete RPMI supplemented with 200ng/ml recombinant human granulocyte macrophage colony stimulating factor (GMCSF) and 100ng/ml recombinant IL-4 for 6 days [32].

## Macrophage and dendritic cell stimulation with SLA

PBMC derived macrophages/dendritic cells were stimulated with SLA at a concentration of 50ug/ml. Uninfected cells and Con–A (50Ug/ml) were used as negative and positive controls respectively. Macrophage cell culture supernatants were harvested at 24h, 48h and 72h time intervals and stored at -20˚C for analysis of cytokines and NO levels. Dendritic cells were

detached using ice cold PBS/EDTA solution at 24h, 48h and 72h time intervals and used for flowcytometric analysis.

## Assay of cytokine and nitric oxide production by macrophages

Supernatants of macrophage cultures from LCL patients, endemic and non-endemic controls were assayed for TNF-α, TGF-β & IL-10 after 24, 48 and 72 hours of stimulation by ELISA according to manufacturer's instructions (R&D Systems, USA). The sensitivity of the cytokine assays were 1.6pg/ml (TNF-α), 4.61pg/ml (TGF-β) and 3.9 pg/ml (IL-10). Nitrite ($NO_2^-$) accumulation in the cell culture supernatants from all the subjects were measured as an indicator of NO production using a standard Griess reagent kit according to manufacturer's instructions (Molecular Probes, USA; minimum detection level 1.0 μM). All assays were performed in duplicate.

## Analysis of expression of cell surface markers on macrophages

Expression of selected cell surface receptors on macrophages was assayed by flow cytometry. After washing with cold PBS, cells were incubated at 4°C for 20 minutes with either CD14-PE (phycoerythrin), CD68-FITC (fluorescein), CD80-APC (allophycocyanin) and CD163-Per-CP (peridinin chlorophyll) or CD 14-PE, CD40-FITC and CD86- Per-CP for staining, washed again with cold PBS, resuspended in cold PBS, acquired immediately and analyzed.

## Assay of cytokine production by dendritic cells

Dendritic cells were analysed for cytokine production by intracellular cytokine staining. In brief, Cells washed twice with cold PBS, stained for specific surface markers (FITC-conjugated anti-CD11c), fixed (fixation buffer, Bio Legend, USA) and permeabilized (10% permeabilization wash buffer, BioLegend, USA). Cells were then stained for intracellular cytokines with APC-conjugated anti-IL-12p70, and PE-conjugated anti-IL-10. Excess antibodies were washed off through two washing steps and the final pellet was dislodged in 2% formaldehyde and acquired by flow cytometry.

## Antibodies and flow cytometry

Single stained cells and unstained cells were used to confirm the specificity of the binding of the antibody of interest and to rule out non-specific receptor binding. All antibodies were purchased from BioLegend, USA. Flowcytometric data was acquired with Cube 8 (Partech) flow cytometer and analyzed using FCS Express 4 (De Novo) software.

## Statistical analysis

Levels of inflammatory biomarkers are presented as medians and interquartile ranges (IQRs). Differences in these biomarker levels between the patients and the control groups were assessed using non-parametric Mann–Whitney U test or Kruskal-Wallis test followed by Dunn's multiple comparison test, as appropriate. Receiver Operating Characteristic (ROC) curve analysis was conducted and area under the curve (AUC) was compared to determine the ability of the inflammatory markers, individually and in combination, to differentiate patients from controls. Analysis was performed using IBM SPSS version 20 and GraphPad Prism version 7 (GraphPad Software, Inc., La Jolla, CA, USA). All statistical tests were two-tailed and a probability (P) value of less than 0.05 was considered statistically significant.

## Results

### Characteristics of the study population

The study population altogether consisted of 60 patients and 90 controls. The socio demographic and clinical characteristics of the patients are summarized in Table 1. Patients consisted of 40 (66.7) males and 20 (33.3) females with a median age was 47 years (range 19–60 years). A majority of the lesions were either papules or nodules (37/60; 61%) with other lesion types seen in fewer numbers (Table 1). None of the patients had similar lesions in the past. All except nine patients had single lesions (51/60). None of the patients showed evidence of visceralization.

### Patient derived macrophages produced a mixed inflammatory pattern after *in vitro* stimulation with SLA

We determined the temporal immune response of patient and control derived macrophages through the comparison of the levels of selected cytokines and NO in cell culture supernatants at 24, 48 and 72 hours (Fig 2). We observed a relative increase in production of TNF-$\alpha$ by patient derived cells which was significantly higher at 48 and 72 hours (when compared with both endemic and non-endemic controls; $p < 0.01$).

Whereas IL-10 secretion by patient derived cells showed a marked increase at 72hrs when compared to both endemic and non-endemic controls ($p < 0.01$), both IL-10 and TNF-$\alpha$ levels in control groups remained relatively low throughout.

Secretion of TGF-$\beta$ by patient derived cells was higher compared to control derived cells at all time points with a sharp rise in levels observed at 48hrs which persisted at 72hrs. The differences in cytokine levels were significant at all 24h (vs endemic controls, $p = 0.027$; vs non-endemic controls, $p = 0.003$), 48h (vs endemic controls, $p < 0.01$; vs non-endemic controls, $p = 0.001$) and 72h (vs endemic controls, $p = 0.008$; vs non-endemic controls, $p < 0.01$) time points.

**Table 1. Sociodemographic and clinical characteristics of patients enrolled in the study.**

|  | Patients with LCL (n = 60) |
|---|---|
| **Age** (years, median ± SD) | 47 ± 13.35 |
| **Male/Female** (number, %) | 40 (66.7)/20(33.3) |
| **Months since onset of lesion** (median, range) | 4 (3–15) |
| **Site of lesion**(number, %) |  |
| Hand | 27 (45) |
| Head | 12 (20) |
| Back | 10 (16.7) |
| Leg | 8 (13.3) |
| Face | 3 (5) |
| **Type of lesion**(number, %) |  |
| Papule | 19 (31.7) |
| Nodule | 18 (30) |
| Plaque | 10 (16.7) |
| Ulcerated nodule | 7 (11.6) |
| Ulcer | 6 (10) |
| **Number of lesions**(number, %) |  |
| Single | 51 (85) |
| Multiple | 9 (15) |

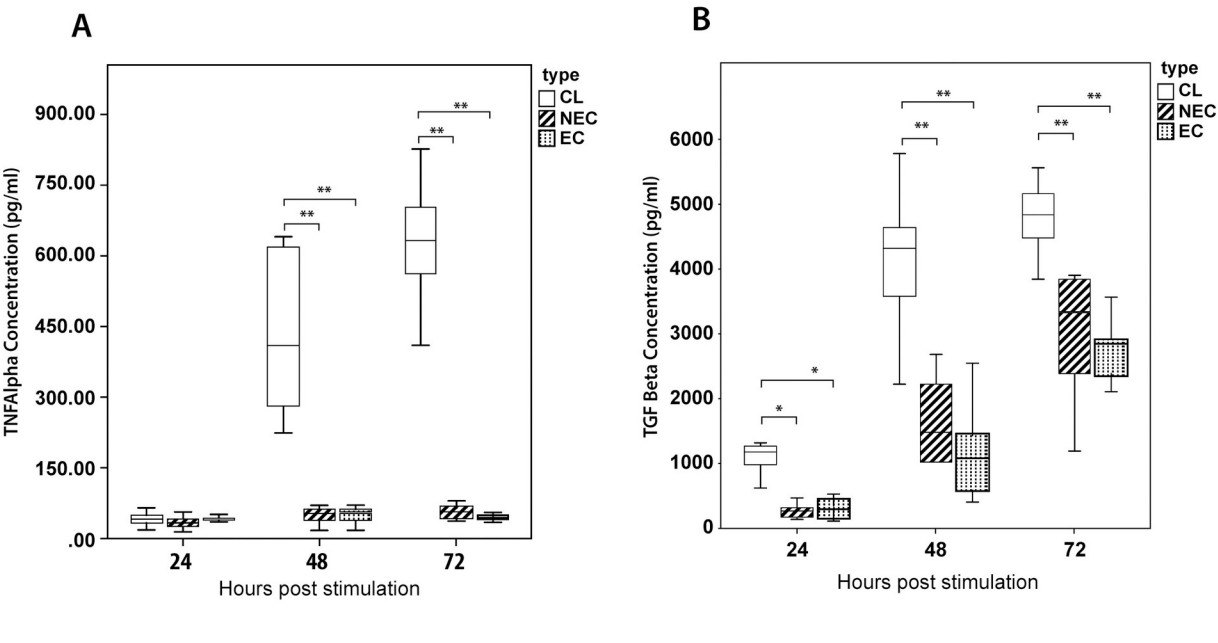

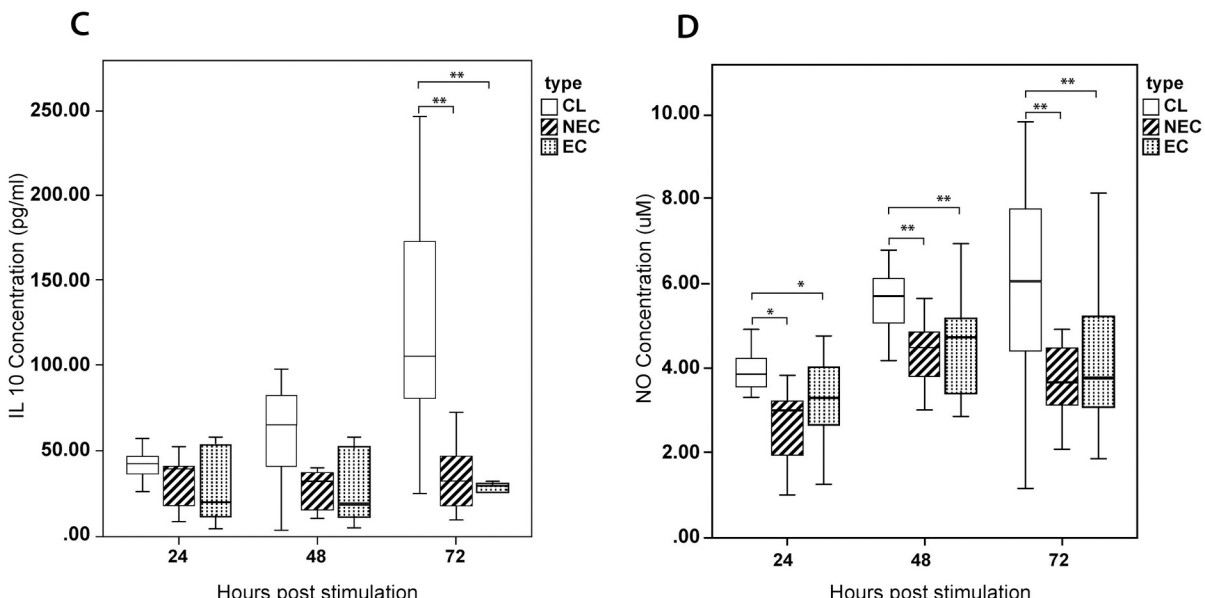

**Fig 2. Comparison of selected inflammatory markers secreted by PBMC derived macrophages stimulated with SLA.** Monocyte derived-macrophages from patients with LCL (n = 20), healthy non endemic controls (n = 10) and healthy endemic controls (n = 20) were stimulated with *L. donovani* antigen (50μg/ml). The supernatant was harvested at 24, 48 & 72 hours and (A) TNF-α (B) TGF-β (C) IL-10 and (D) NO levels were determined by ELISA and Griess test respectively (in duplicate). The box represents the IQR (i.e. the middle 50% of the observations) with the horizontal line representing the median. The whiskers represent the main body of the data, indicating the range of the data. For statistical analysis, nonparametric Kruskal-Wallis test followed by Dunn's multiple comparison test was used (*p<0.05; **p<0.01; ***p<0.001).

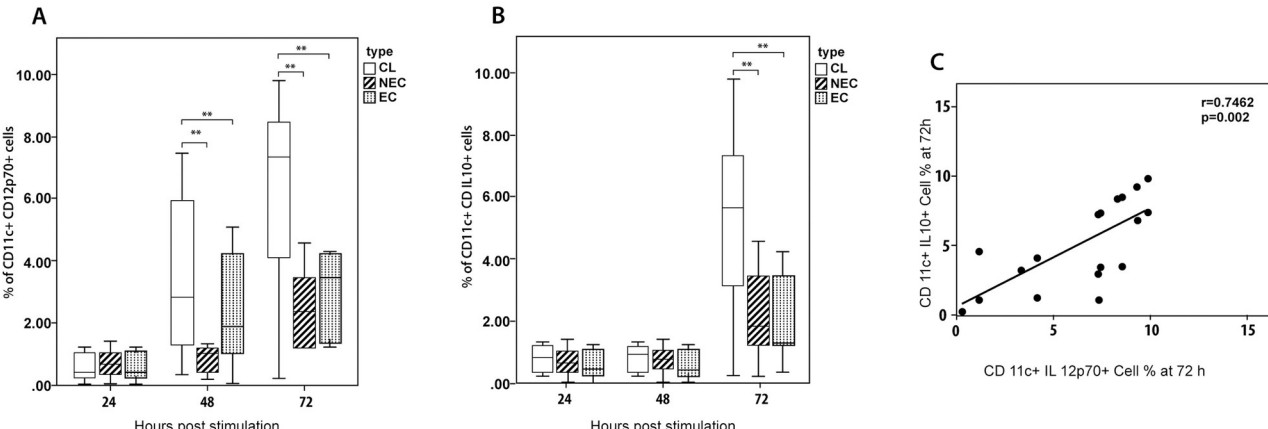

**Fig 3. Comparison of selected cytokines production by PBMC derived dendritic cells stimulated with SLA.** Dendritic cells from LCL (n = 20), healthy non endemic controls (n = 10) and healthy endemic controls (n = 20) were stimulated with *L. donovani* antigen (50μg/ml). The supernatant was harvested at 24, 48 and 72 hours and intracellular (A) IL-12p70 and (B) IL-10 levels were determined by flowcytometry(in duplicate). The box represents the IQR (i.e. the middle 50% of the observations) with the horizontal line representing the median. The whiskers represent the main body of the data, indicating the range of the data. Cytokine levels were compared using nonparametric Kruskal-Wallis test followed by Dunn's multiple comparison test ($^*p{<}0.05$; $^{**}p{<}0.01$; $^{***}p{<}0.001$). (C) IL-12p70 and IL-10 levels showed a strong positive correlation at 72 h (Spearman correlation test, $p{<} 0.05$).

Nitric oxide (NO) levels showed a pattern of differences similar to TGF-β with secretion by patient derived cells being higher at 24h (vs endemic controls, p = 0.03; vs non-endemic controls, p = 0.009), 48h (vs endemic controls, p = 0.002; vs non-endemic controls, p<0.01) and 72h time points (vs endemic controls, p = 0.038; vs non-endemic controls, p = 0.009).

None of the cytokine combinations showed a strong correlation of secretion levels at any of the three time points (https://figshare.com/s/6d7203ba834cea22639c).

## Patient derived dendritic cells produced a mixed inflammatory pattern after *in vitro* stimulation with SLA

A similar temporal analysis was carried out on the immune response by dendritic cells derived from patients and controls after in vitro stimulation with SLA (Fig 3A & 3B). IL-10 and IL-12p70 intra cellular markers were used in combination with CD11c to determine the intracellular cytokine production by dendritic cells.

The percentage of CD11c$^+$IL-12p70$^+$ in patient derived cells showed a steady increase over time unlike the cells derived from controls and this increase was significant at both 48h (vs endemic controls, p = 0.026; vs non-endemic controls, p = 0.027) and 72h (vs endemic controls, p = 0.042; vs non-endemic controls, p = 0.01) time points. The percentage of CD11c$^+$IL10$^+$ cells showed a marked rise in patient derived cells at 72 hours and was significantly higher compared to cells derived from both endemic (p = 0.018) and non-endemic (p = 0.027) controls.

IL-12p70 and IL-10 levels showed a strong positive correlation at 72 h (Fig 3C) (r = 0.7462, p = 0.0002).

## Pro inflammatory vs. regulatory cytokines and utility as diagnostic biomarkers

The temporal dynamics of inflammatory mediators produced by patient derived cells showed a distinct pattern of early responses to parasite antigen (Fig 4A–4F). We further analysed the

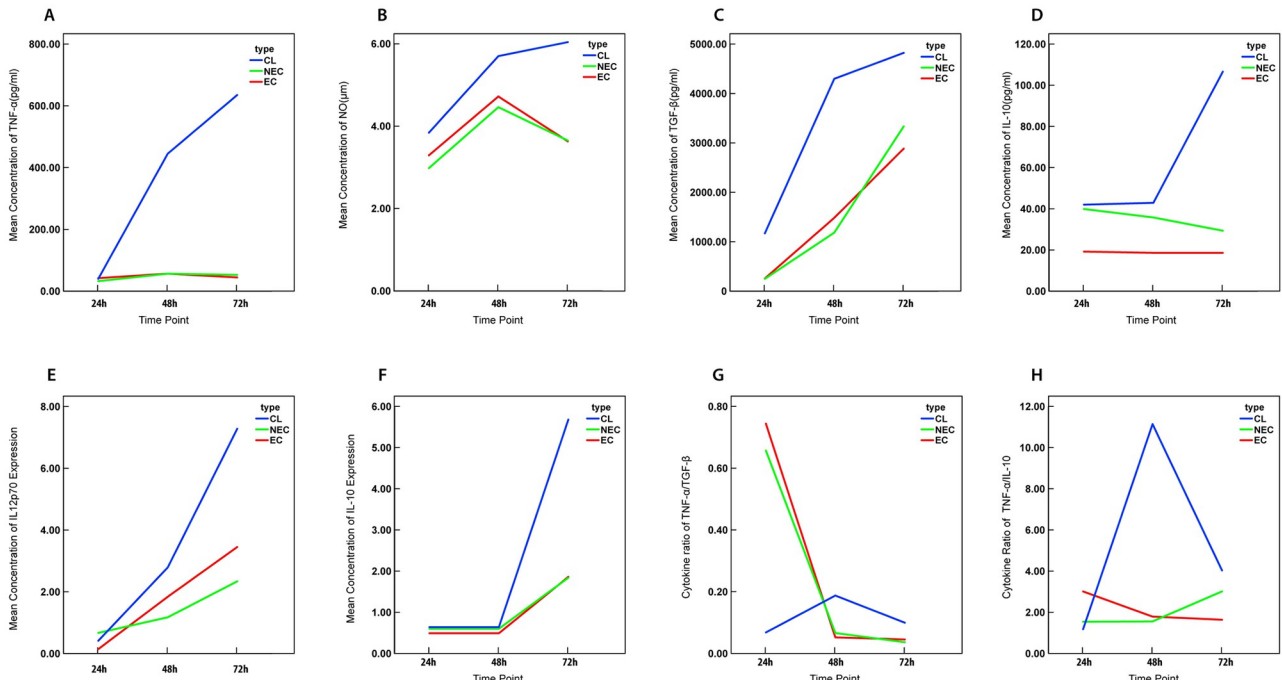

**Fig 4. Change in mean concentration of inflammatory markers and their ratios over time.** Line graphs representing the mean production of (A) TNF-α (B)NO (C)TGF-β (D) IL-10 in macrophages and (E)IL-12p70 (F)IL-10 in dendritic cells stimulated with *L. donovani* antigen (50μg/ml) and the ratio of the mean concentration of (G)TNF-α /TGF-β (H)TNF-α/ IL-10 at 24, 48 and 72 hour time intervals.

cytokine responses at 72hrs, the time point at which most of the significant differences were observed. Macrophage derived pro-inflammatory to anti-inflammatory cytokine ratios were seen to be significantly higher in patients compared to controls (TNF-α to IL-10, p = 0.008; TNF-α to TGF-β, p< 0.001) (Fig 4G–4H). While the proportion of dendritic cells producing IL-12p70 and IL-10 also favoured a pro inflammatory response as indicated by IL-12p70 /IL-10 ratio, the differences between the patients and the controls was not significant.

We performed receiver operating characteristic (ROC) curve analysis in order to assess the predictive value of the levels of the inflammatory markers to accurately identify patients with active disease. In line with the profile of significant differences between patients and controls, the ROC curves showed best discrimination at 72 hours for all 3 cytokines (TNF-α, IL-10 and TGF-β) produced by macrophages (Table 2) (Fig 5).

The endemic and non-endemic controls were pooled for the above analyses since significant differences were not observed in cytokine levels between these two groups of controls.

### Patient derived macrophages displayed M1 polarization after stimulation with SLA

Macrophage polarization into M1or M2 subsets was identified by expression of CD14, CD68, CD80 and CD163 markers. CD14 and CD68 markers are expressed by monocytes and macrophages and M1 and M2 cells are known to predominantly express CD80 and CD163 respectively. The frequency of CD14+CD68+CD80+ macrophages were significantly higher in LCL patients when compared to healthy individuals at 48h (vs endemic controls, p = 0.012; vs non-endemic controls, p = 0.013) and 72h (vs endemic controls, p = 0.045; vs non-endemic controls, p = 0.006) time points (Fig 6B and 6C). But in contrast the fraction of CD14+CD68+

**Table 2. Performance of tested analytes as biomarkers to identify active disease status.**

| Marker | AUC (95% CI) | p value | SE | SP | Optimum cut off |
|---|---|---|---|---|---|
| TNF-α (pg/ml) | 1 (1.00–1.00) | <0.001 | 1.000 | 1.000 | 177.48 |
| IL-10 (pg/ml) | 1(1.00–1.00) | <0.001 | 1.000 | 1.000 | 81.13 |
| TGF-β (pg/ml) | 0.957 (0.901–1.00) | <0.001 | 0.95 | 0.80 | 3834.55 |

AUC, area under the curve; SE, sensitivity; SP, specificity; CI, confidence interval

CD163⁺ cells were low in the same subjects at all the time points (https://figshare.com/s/dbd757843c59ae7811bc) suggesting a M1 polarization, which is associated with pro-inflammatory microbicidal responses.

### *Leishmania donovani* antigen promotes early activation but not maturation of patient derived macrophages

CD40⁺ and CD86⁺ cells were gated on CD14⁺ cells in order to determine the cell activation and maturation respectively of macrophages upon stimulation with SLA. CD14⁺CD40⁺ cell population was significantly higher at the 24h time point in LCL patients than the controls (vs

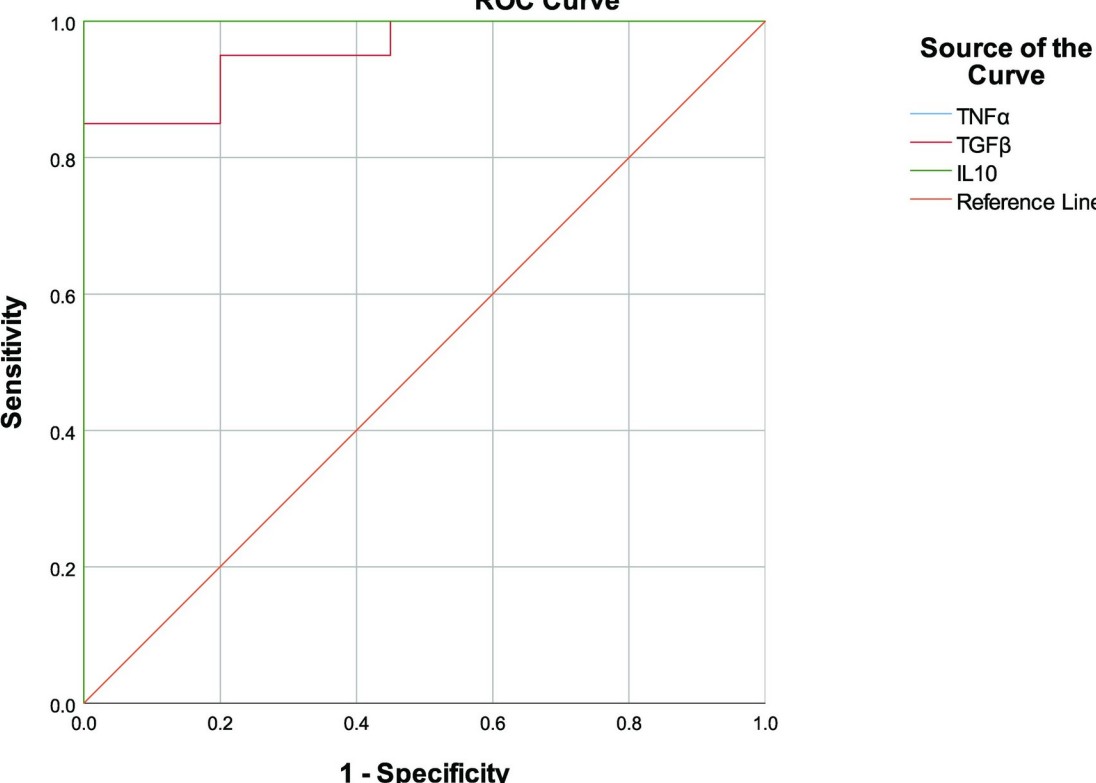

**Fig 5. ROC curves for inflammatory markers produced by macrophages.** ROC curves were calculated for TNF-α, NO, TGF-β and IL-10 produced by macrophages which showed significant differences in mean concentration levels in the culture supernatant. Detailed information on the AUCs is shown in Table 2. Higher values on the y-axis correspond to higher sensitivity, whereas lower values on the x-axis correspond to higher specificity.

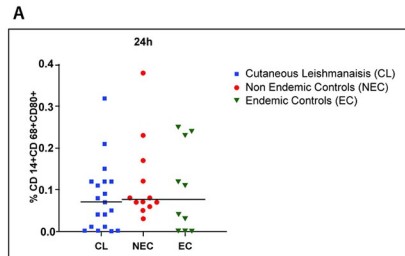
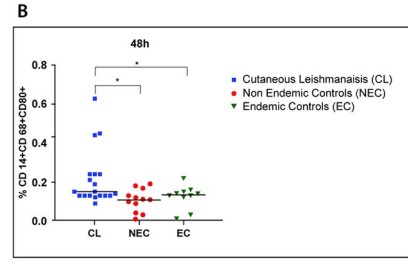
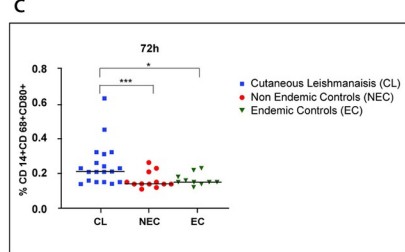

**Fig 6. Distribution of M1 macrophages in patients and controls following stimulation with SLA.** Monocyte derived-macrophages from LCL (n = 20), healthy non endemic controls (n = 10) and healthy endemic controls (n = 20) were stimulated with *L.donovani* antigen (50μg/ml). Expression of CD14, CD68 and CD80 were analysed by flowcytometry at A) 24h, B) 48h and C) 72h to determine M1 polarization (in duplicate). In the vertical scatter plot each symbol represents a different subject. Horizontal bars represent the median. For statistical analysis, nonparametric Kruskal-Wallis test followed the Dunn's multiple comparison test was used was used (*p<0.05; **p<0.01; ***p<0.001).

endemic controls, p = 0.006; vs non-endemic controls, p = 0.011) suggesting early activation of the monocyte/macrophage system in this group (Fig 7A) with comparable levels of activation of patient and control derived cells at 48 and 72 hours (Fig 7B and 7C). However, the CD14$^+$CD86$^+$ cell population did not differ between the patients and the controls throughout the time course. (https://figshare.com/s/04b6424f668c0ca4cafe).

## TGF-β secretion by macrophages is associated with severity of the cutaneous lesions

The immune response was evaluated in relation to lesion characteristics with late stage (lesion present for more than six months) and/or ulcerated lesions considered as features of severity. Cells from patients with both acute and chronic lesions consistently showed a sharp increase in production of TGF-β at 48 and 72hrs, with those from late stage lesions showing significantly higher levels at 48hrs (p = 0.041) (Fig 8A).

Comparative evaluation with lesion type showed a similar trend with cells from patients with ulcerated lesions showing significantly higher levels of TGF-β at 48hrs (p = 0.009) (Fig 8B). IL-10, TNF-α and NO levels did not differ significantly with the duration of the lesion or presence of ulceration. A similar analysis of IL-10 and IL12p70 secreted by dendritic cells also did not show any significant differences (https://figshare.com/s/04b6424f668c0ca4cafe).

The same pattern of MI polarization was retained in acute lesions (S1A Fig). While cells from patients with chronic lesions showed a marked increase in M2 macrophages at the same

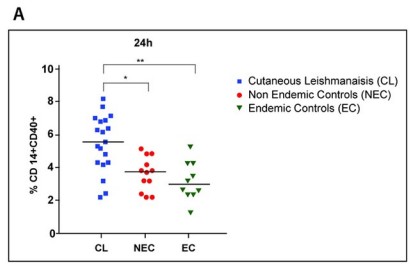
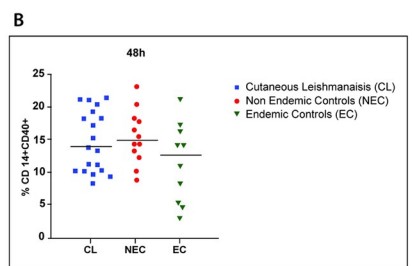
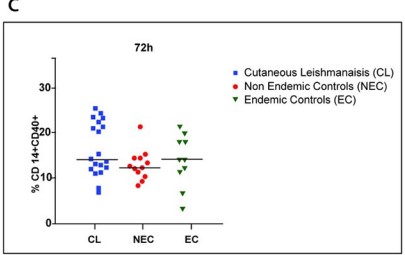

**Fig 7. Distribution of activated macrophages over time in patients and controls following stimulation with SLA.** Monocyte derived macrophages from LCL (n = 20), healthy non endemic controls (n = 10) and healthy endemic controls (n = 20) were stimulated with *L.donovani* antigen (50μg/ml). Percentage of activated cells was determined at A) 24h, B) 48h and C) 72h by analyzing expression of CD14 and CD40 by flowcytometry. In the vertical scatter plot each symbol represents a different subject. Horizontal bars represent the median. For statistical analysis, nonparametric Kruskal-Wallis test followed the Dunn's multiple comparison test was used was used (*p<0.05; **p<0.01; ***p<0.001).

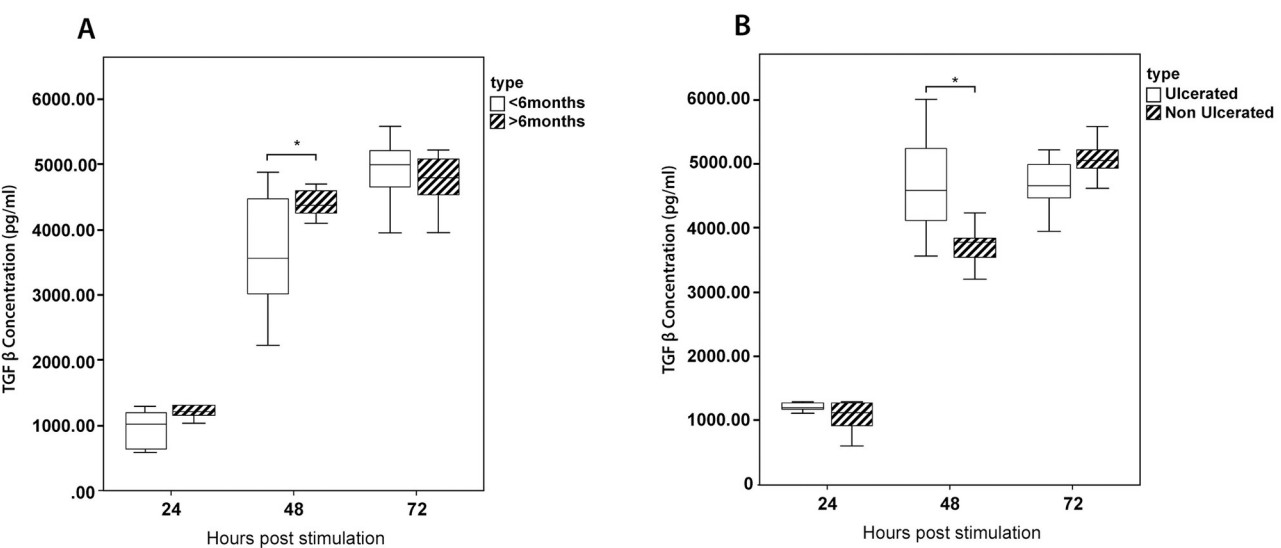

**Fig 8. Kinetics of TGF-β in relation to lesion type and duration.** The production of TGF- β was compared between patients with (A) acute (lesions less than six months/chronic lesions (lesions more than 6 months) and (B) ulcerated/non ulcerated lesions. The box represents the IQR (i.e. the middle 50% of the observations) with the horizontal line representing the median. The whiskers represent the main body of the data, indicating the range of the data. For statistical analysis, nonparametric Kruskal-Wallis test followed by the Dunn's multiple comparison test was used (*p<0.05; **p<0.01; ***p<0.001).

time point when compared to acute lesions, this difference was not statistically significant (p>0.05) (S1B Fig). The presence of activated macrophages also appeared to favour ulcerated lesions (S2 Fig).

## Discussion

A role for cytokines and phenotypic cell profiles is well recognized in contributing to clinical presentations and progression of leishmaniasis. We focused on cutaneous leishmaniasis caused by *L. donovani* that is endemic in our setting, to study the host cytokine and cellular responses across different time points. While being an atypical host-parasite combination relative to global epidemiological patterns, the remarkable uniformity in presentation in the local setting presented a unique opportunity to study the immune responses which are likely key contributors in determining a cutaneous phenotype.

In this study, patient derived cells showed higher levels of secretion of all the inflammatory markers studied (TNF-α, IL-10, TGF-β, IL-12p70, NO) when stimulated with antigen derived from local parasite isolates. The fluctuations in cytokine levels appeared to dampen and stabilize about three days post infection with distinct inflammatory patterns in patients compared to controls. This was further supported by the ROC analysis of the inflammatory markers, with macrophage derived TNF-α, IL-10 and TGF-β showing a clear differentiation between patients and controls at this time point.

We investigated several pro inflammatory markers in this study; TNF-α, NO and IL-12p70. TNF-α plays a crucial role in eliminating intra macrophage parasites through sustained induction of nitric oxide synthase and also promotes development of a protective Th1/IFN-γ response [33]. While NO has been demonstrated to facilitate parasite killing in CL caused by different species of *Leishmania* [34, 35] both TNF-α and NO have been implicated in excessive inflammatory reactions which result in features such as ulceration [36]. IL-12 acts as a vital cytokine which bridges the innate and adaptive immune arms and

promotes development of a protective Th1 response [37]. The patient derived cells in our study consistently produced higher levels of NO while TNF-α and IL-12p70 levels were higher at 48 and 72hrs. In a similar study, PBMC derived macrophages from patients with CL and MCL due to *L. braziliensis* demonstrated higher values of TNF-α at 48 hours after infection [30], while another focusing on the same parasite species demonstrated a predominance of TNF-α during active disease [38]. Nitric oxide levels have also shown to be higher in CL by some investigators [36] while others reported no changes [30]. Interestingly, a study on J774-macrophage cell line has shown SLA derived by both *L. tropica* and *L. major* causing CL in Morocco to significantly inhibit the production of NO in a dose dependent manner [39]. Priming of dendritic cells to secrete IL-12 has been observed in vitro with *L. major* which causes LCL [40], while *L. tropica* which causes more persistent cutaneous lesions did not induce similar responses [32]. Experimental studies have also supported the species dependent differential regulation of the immune responses with reports of reduced number of differentiated monocyte derived dendritic cells present in lesions due to *L. mexicana* compared to *L. major* [41]. Selected anti-inflammatory markers, IL-10 and TGF-β by macrophages and IL-10 by dendritic cells were also evaluated in this study. IL-10 is a well-known cytokine to exert suppressive biological effects on many cell types, including inhibition of macrophage mediated Th1 cell activation [42, 43]. TGF-β is an anti-inflammatory cytokine which predisposes to disease progression and severe manifestations as reported in a variety of infective and inflammatory conditions. In our study IL-10 secreted by both macrophages and dendritic cells were higher in patient derived cells at 72hrs while TGF-β was higher at all three time points. Higher expression of IL-10 in CL and MCL is reported in other studies [44]. A study on role of TGF-β in CL caused by *Leishmania amazonensis*, *L. donovani chagasi* and *L. braziliensis* has shown that TGF-β led to an increase in parasite numbers when added to in vitro culture [45]. Our results of the comparison of early and late lesions as well as progression to ulceration suggests TGF-β as a key modulator in severity of the lesions. Another study has also demonstrated that TGF-β is associated with chronic forms of the disease [46, 47] or long lasting atypical lesions [48]. It is widely accepted that ulcerated lesions produce more anti-inflammatory cytokines which delays the healing [49].

Overall, our findings showed a mixed inflammatory pattern to be characteristic of the pathogenic process which results in LCL that we observe in the country. These findings suggest that the effects of anti-inflammatory cytokines which would promote progression of the lesions, is balanced by the pro-inflammatory cytokine responses as well as a sufficient oxidative burst, resulting in limited parasite spread and well localized lesions. Mixed cytokine profiles have also been reported by investigators who conducted similar studies on CL caused by other species [50, 51]. A positive correlation between pro and anti-inflammatory mediators as we observed in dendritic cell responses also are in favour of a regulated response which kills the parasite while limiting tissue injury and is in agreement with reports on similar investigations [52–54]. While in endemic settings the diagnosis of CL is often clinical, aided by microscopy, treated lesions with a poor response or atypical presentations may benefit from complimentary tests based on other biomarkers.

The two groups of controls in this study showed similar cytokine dynamics. While the endemic controls were assumed to have been exposed to infected sandfly bites we did not confirm such exposure by a Montenegro skin test (delayed hypersensitivity testing) primarily due to non-availability of a suitable standardized antigen preparation as well as logistical constraints. Thus, absence or low degree of 'actual' exposure could be a contributory factor for this similarity unlike the intermediate profiles reported by others who investigated asymptomatic infections [55, 56]. Another limitation of this study was the relatively small sample size and especially the characteristics of the immune profile of the subgroups with chronic or

ulcerated lesions, should be verified in a larger population. The ROC analysis was not extended to differentiating these more severe clinical presentations due to the same reason.

Whereas in vitro studies allow easy manipulation of cells of the immune system and thus simultaneous study of a variety of immune markers, this may not mimic all the conditions *in situ*, in cutaneous lesions. However, our study findings were similar to others who also reported differences in cytokines such as TNF-α, IL-12p70 and IL-10 between local patients and controls by study of skin biopsies [25, 26]. The clinical picture is further supported by cytokine ratios which indicated a greater influence of the pro-inflammatory cytokines. It is also apparent from our results on temporal profiles of expression of cell surface receptors that *Leishmania* parasites cause early activation of macrophages and polarization towards a M1 phenotype contributing to limiting the spread of the parasites. In contrast to a mixed picture as we observed in LCL in the present study, in post-kala-azar dermal leishmaniasis (PKDL), a cutaneous sequel to *L. donovani* infections which cause visceral leishmaniasis, predominance of anti-inflammatory cytokines with M2 polarization of macrophages has been reported [57]. Polarizing of activated macrophages is influenced by the milieu of cytokines, growth factors and microbial products in its micro environment [58]. Among the range of cytokines produced by M1 polarized macrophages, primarily IL-12 induces Th1 cells while pro-inflammatory cytokines such as TNFα and IL-6, production of reactive oxygen intermediates (ROI) and nitric oxide synthase-2 (NOS-2/iNOS)-dependent reactive nitrogen intermediates (RNI) characterizes the microbicidal activity. IL-10 and TGF-β which are mainly associated with alternatively/M2 polarized macrophages induce Th2/Treg cell differentiation promoting parasite spread [11].

The fact that the infecting parasite species/strain may modulate the immune response in leishmaniasis has been suggested by many experimental and human studies [32, 59]. Interestingly a recent experimental study of CL by *L. major* has shown macrophage differentiation into different functional types in a strain dependent manner in the absence of lymphocytes, thus with direct implications on the innate response elicited [60]. It is likely that the local parasite also has such antigenic differences when compared to the typical isolates of the species, which has resulted in altered pathogenicity. While previous analyses of genomic sequences by us [61] and others [62] also suggest such differences, a comparison of immune profiles of local patients with LCL and VL would provide more confirmatory evidence on the differential immune responses which influence the phenotype.

In the physiological status, the effects of a cytokine would depend not only on secretory levels but on an interacting network with other cytokines as well as a spectrum of other inflammatory mediators. Effects of other key cytokines in the innate response such as IFN-γ as well as activity of T cells, including cytokine responses of Th17 and T regulatory cells in addition to conventional Th1 and Th2 cells, would need to be explored to obtain a better understanding of the inflammatory dynamics concerned. The early interactions at the host-parasite interface and changes in mediating molecules such as toll-like-receptors (TLRs) [63, 64] are some other factors to be considered in discerning the pathogenic mechanisms.

## Conclusions

Collectively, our data shows that a time dependent mixed inflammatory pattern but with a high pro-inflammatory to regulatory cytokine ratio with classically activated macrophages underlies the clinical phenotype of LCL caused by a dermotropic variant of *L. donovani*. Similar observations of mixed patterns in LCL being reported from different geographical foci suggest that some of the pathogenic mechanisms underlying LCL are conserved across species, thus making these suitable candidates for immune modulatory or immune prophylactic

interventions. Our data provides groundwork to evaluate suitability of developing assays based on such serological biomarkers in assessing disease status and prognosis. Further, the study adds to the information on how this intra cellular parasite modulates first line host defenses and provides evidence for likely host factors which prevent visceralization of the parasite.

## Supporting information

**S1 Fig. Comparison of M1/M2 polarization of macrophages derived from patients with acute and chronic lesions.** A) MI and B) M2) polarization of macrophages stimulated with SLA was compared between patients with acute (duration less than six months) and chronic lesions. The box represents the IQR (i.e. the middle 50% of the observations) with the horizontal line representing the median. The whiskers represent the main body of the data, indicating the range of the data. For statistical analysis, nonparametric Kruskal-Wallis test followed by the Dunn's multiple comparison test was used ($^*p < 0.05$; $^{**}p < 0.01$; $^{***}p < 0.001$).
(TIF)

**S2 Fig. Comparison of distribution of activated macrophages derived from patients with ulcerated and non ulcerated lesions.** The distribution of activated macrophages following stimulation with SLA was compared between patients with ulcerated and non ulcerated lesions. The box represents the IQR (i.e. the middle 50% of the observations) with the horizontal line representing the median. The whiskers represent the main body of the data, indicating the range of the data. For statistical analysis, nonparametric Kruskal-Wallis test followed by the Dunn's multiple comparison test was used ($^*p < 0.05$; $^{**}p < 0.01$; $^{***}p < 0.001$).
(TIF)

## Acknowledgments

The authors wish to thank Dr. KKVN Somarathna, Dr. L. Pathiraja, staff members of the Dermatology Clinics of General Hospitals Anuradhapura, Hambanthota and Padaviya and Dr. Nuwani Manamperi for assistance with patient recruitment, Prof. Neelika Malawige of the Department of Immunology and Molecular Medicine, Faculty of Medical Sciences, University of Sri Jayawardenapura for guidance and facilitation of flowcytometry studies and Mr. Sudath Weerasingha for technical assistance.

## Author Contributions

**Conceptualization:** Nilakshi Samaranayake, Nadira D. Karunaweera.

**Data curation:** Hiruni Wijesooriya.

**Formal analysis:** Hiruni Wijesooriya.

**Funding acquisition:** Nadira D. Karunaweera.

**Investigation:** Hiruni Wijesooriya, Nilakshi Samaranayake.

**Methodology:** Hiruni Wijesooriya.

**Project administration:** Nadira D. Karunaweera.

**Resources:** Nilakshi Samaranayake, Nadira D. Karunaweera.

**Supervision:** Nilakshi Samaranayake, Nadira D. Karunaweera.

**Validation:** Nilakshi Samaranayake, Nadira D. Karunaweera.

**Writing – original draft:** Hiruni Wijesooriya.

**Writing – review & editing:** Nilakshi Samaranayake, Nadira D. Karunaweera.

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
