## [Decision Letter · Decision Letter 0]

22 Aug 2022

PONE-D-22-17113Cytokine and phenotypic cell profiles in human cutaneous leishmaniasis caused by Leishmania donovaniPLOS ONE

Dear Dr. Nilakshi Samaranayake,

Thank you for submitting your manuscript to PLOS ONE. After careful consideration, we feel that it has merit but does not fully meet PLOS ONE’s publication criteria as it currently stands. Therefore, we invite you to submit a revised version of the manuscript that addresses the points raised during the review process.

We look forward to receiving your revised manuscript.

Kind regards,

Alireza Badirzadeh

Academic Editor

PLOS ONE

Journal Requirements:

“This work was supported by the National Institute of Allergy and Infectious Diseases of the National Institutes of Health, USA, under award number U01AI136033 to NK. The content is solely the responsibility of the authors and does not necessarily represent the official views of the National Institutes of Health.”

Reviewers' comments:

Reviewer's Responses to Questions

**Comments to the Author**

1. Is the manuscript technically sound, and do the data support the conclusions?

Reviewer #1: No

Reviewer #2: Yes

Reviewer #3: Yes

2. Has the statistical analysis been performed appropriately and rigorously? 

Reviewer #1: Yes

Reviewer #2: Yes

Reviewer #3: I Don't Know

3. Have the authors made all data underlying the findings in their manuscript fully available?

Reviewer #1: Yes

Reviewer #2: Yes

Reviewer #3: Yes

4. Is the manuscript presented in an intelligible fashion and written in standard English?

Reviewer #1: Yes

Reviewer #2: Yes

Reviewer #3: Yes

5. Review Comments to the Author

Reviewer #1: The manuscript presents a very nice concept that is innate profile of the immune response that mitigates the outcome of l. donovani infection in Seri Lanka however some points are not clear for the reviewer which are summarized below:

1- The macrophage and Dendritic cells protocols are much different from conventional standard protocols. The authors need to characterize each differentiated cell type by flow cytometry before any experiment based on these cell types could be further evaluated.

2- CD11c alone is not enough for mature human DC characterization.

3- You have not tested the previous exposure history in endemic controls which is necessary to be tested.

4 It is not clear why do macrophages derived from monocytes should produce Nitric oxide in the absence of any IFN-g cytokine? the same way why do authors expect IL-12 production from DCs in the absence of any stimulatory cytokine fro DC? is SLA stimulation potent enough to direct IL-12 expression without any inducing signals from PRRS? These results were much more realistic if designed with whole live parasite infection.

5- It is not clear why monocyte derived macrophages should turn into M1 macrophages after SLA stimulation?

6- Finally the whole discussion must compare the results from this study with the same studies based on MoDC or MoMQ stimulated with parasite antigens.

Reviewer #2: Dear Editor

In this manuscript, the authors aim to characterize the innate cellular immune responses associated with locally acquired cutaneous leishmaniasis due to L. donovani. They hypothesized that distinct profiles of the early immune response determined the clinical phenotype of localized cutaneous leishmaniasis observed in Sri Lanka. The introduction, M & M, and discussion sections need to be revised according my comments. After close review, I recommended that the manuscript is not accepted in the present format. There are several concerns that must be addressed:

- Please add new references in the introduction sections. Also add a paragraph about all causative agents of CL in the world such as L. major, L. tropica, and so on. In addition, please add more details about Th1/Th2 and its mechanisms in CL.

Use the following articles:

1- "Arginase activity of Leishmania isolated from patients with cutaneous leishmaniasis." Parasite immunology 39.9 (2017): e12454.

2- "Case report: First coinfection report of mixed Leishmania infantum/Leishmania major and human immunodeficiency virus–acquired immune deficiency syndrome: report of a case of disseminated cutaneous leishmaniasis in Iran." The American journal of tropical medicine and hygiene 98.1 (2018): 122.

3- "First case report of atypical disseminated cutaneous leishmaniasis in an opium abuser in Iran." Revista do Instituto de Medicina Tropical de São Paulo 60 (2018).

4-. "Sambucus ebulus extract stimulates cellular responses in cutaneous leishmaniasis." Parasite Immunology 41.1 (2019): e12605.

5- "Super infection of cutaneous leishmaniasis caused by Leishmania major and L. Tropica to Crithidia fasciculata in Shiraz, Iran." Iranian Journal of Public Health 48.12 (2019): 2285.

6- "Arginase/nitric oxide modifications using live non-pathogenic Leishmania tarentolae as an effective delivery system inside the mammalian macrophages." Journal of Parasitic Diseases 45.1 (2021): 65-71.

7- "Immunogenic properties of empty pcDNA3 plasmid against zoonotic cutaneous leishmaniasis in mice." Plos one 17.2 (2022): e0263993.

- Please add ethics approval code in the manuscript.

- Line 92: Add complete information regarding the following sentences “Department of Parasitology, Faculty of Medicine, University of Colombo” such as city and country.

- Please explain completely the mentioned section: “Preparation of soluble Leishmania antigen (SLA)”.

- Please update your L. donovani parasites cell culture section to reflect the passage number of the cells.

- Which phase of L. donovani did you use for parasite culture? Stationary or logarithmic?

- Pleas add more information for this section “Cytokine and nitric oxide assays” and add the following articles for cytokine and nitric oxide assessment.

"Immunogenic properties of empty pcDNA3 plasmid against zoonotic cutaneous leishmaniasis in mice." Plos one 17.2 (2022): e0263993.

"Arginase/nitric oxide modifications using live non-pathogenic Leishmania tarentolae as an effective delivery system inside the mammalian macrophages." Journal of Parasitic Diseases 45.1 (2021): 65-71.

-In discussion section please add more details about Th1/Th2 and its mechanisms in CL.

Reviewer #3: Although the topic is interesting, it is necessary to revise the text of the article comprehensively.

1- Abbreviations in the abstract and the main text of the article are separate. Some abbreviations have not been repeated even once in the abstract, so some abbreviations are not necessary in abstracts. For example: in line 9 Peripheral blood mononuclear cell (PBMC) and etc.

2- Keywords are not available at the end of the abstract and also the number of keywords is too high. Please remove redundant and duplicate items.

3- What was the reason for choosing cytokines? For example, why didn't you use other cytokines such as IFN-γ, IL-12, which are important in the immune response of leishmaniasis?

4- Please mention the exact time of the study in the materials and methods section in both the abstract and the main text of the article.

5- In line 100, considering the high sensitivity and specificity of the direct agglutination test (DAT), please authors explain why you did not use this test to exclude serological evidence of visceral involvement?

6- Mention the inclusion and exclusion criteria of patients and control group exactly.

7- Has this profile been analyzed and discussed in endemic and non-endemic areas?

8- In line 145 please specify how many times the cytokine assay by using ELISA was repeated? Duplicate or triplicate?

9- Please correct all the captions of the figures and the captions of the tables.

10- Overall the methodology is not well sectioned in the abstract and the main text of the article, and this sectioning should also be included in the results.

6. PLOS authors have the option to publish the peer review history of their article (what does this mean?). If published, this will include your full peer review and any attached files.

Reviewer #1: No

Reviewer #2: No

Reviewer #3: No

---

## [Author Response · Author response to Decision Letter 0]

2 Nov 2022

This is provided in table format attached to the cover letter.

---

## [Decision Letter · Decision Letter 1]

21 Nov 2022

Cytokine and phenotypic cell profiles in human cutaneous leishmaniasis caused by Leishmania donovani

PONE-D-22-17113R1

Dear Dr. Nilakshi Samaranayake,

We’re pleased to inform you that your manuscript has been judged scientifically suitable for publication and will be formally accepted for publication once it meets all outstanding technical requirements.

Kind regards,

Alireza Badirzadeh

Academic Editor

PLOS ONE

Additional Editor Comments (optional):

Reviewers' comments:

Reviewer's Responses to Questions

**Comments to the Author**

1. If the authors have adequately addressed your comments raised in a previous round of review and you feel that this manuscript is now acceptable for publication, you may indicate that here to bypass the “Comments to the Author” section, enter your conflict of interest statement in the “Confidential to Editor” section, and submit your "Accept" recommendation.

Reviewer #3: All comments have been addressed

2. Is the manuscript technically sound, and do the data support the conclusions?

Reviewer #3: Yes

3. Has the statistical analysis been performed appropriately and rigorously? 

Reviewer #3: Yes

4. Have the authors made all data underlying the findings in their manuscript fully available?

Reviewer #3: Yes

5. Is the manuscript presented in an intelligible fashion and written in standard English?

Reviewer #3: Yes

6. Review Comments to the Author

Reviewer #3: (No Response)

7. PLOS authors have the option to publish the peer review history of their article (what does this mean?). If published, this will include your full peer review and any attached files.

Reviewer #3: No

---

## [Editor Report · Acceptance letter]

23 Dec 2022

PONE-D-22-17113R1 

Cytokine and phenotypic cell profiles in human cutaneous leishmaniasis caused by *Leishmania donovani*

Dear Dr. Samaranayake:

I'm pleased to inform you that your manuscript has been deemed suitable for publication in PLOS ONE. Congratulations! Your manuscript is now with our production department. 

Kind regards, 

on behalf of

Dr. Alireza Badirzadeh 

Academic Editor

PLOS ONE